# Genome-Wide Identification and Analysis of Chitinase-Like Gene Family in *Bemisia tabaci* (Hemiptera: Aleyrodidae)

**DOI:** 10.3390/insects12030254

**Published:** 2021-03-17

**Authors:** Zhengke Peng, Jun Ren, Qi Su, Yang Zeng, Lixia Tian, Shaoli Wang, Qingjun Wu, Pei Liang, Wen Xie, Youjun Zhang

**Affiliations:** 1Institute of Vegetables and Flowers, Chinese Academy of Agricultural Sciences, Beijing 100081, China; zkpeng0827@163.com (Z.P.); renjun@caas.cn (J.R.); zengyang73@163.com (Y.Z.); tianlixia555@126.com (L.T.); wangshaoli@caas.cn (S.W.); wuqingjun@caas.cn (Q.W.); xiewen@caas.cn (W.X.); 2Department of Entomology, China Agricultural University, Beijing 100193, China; liangcau@cau.edu.cn; 3Hubei Engineering Technology Center for Pest Forewarning and Management, College of Agriculture, Yangtze University, Jingzhou 434025, China; suqicaas@163.com

**Keywords:** *Bemisia tabaci*, chitinase, glycosyl hydrolase family, phylogenetic analysis, RNA interference

## Abstract

**Simple Summary:**

For the sake of growth and development, old exuviums of *Bemisia tabaci* nymphs should be periodically substituted by new ones until a final emergence turns them into adults. During ecdysis, chitinases are of great importance in chitin degradation and turnover of cuticles, which provides us potential targets for RNA interference-based *B. tabaci* management. Therefore, we annotated 14 chitinase and chitinase-like genes in *B. tabaci* based on data of genome and transcriptomes. With the help of a nanomaterial-promoted RNAi method, we found that silencing of *BtCht10*, *BtCht5,* and *BtCht7* resulted in significant increase of death rate on *B. tabaci* nymphs and the developmental duration was noticeably postponed for *BtCht2*-silenced nymphs.

**Abstract:**

Chitinases are of great importance in chitin degradation and remodeling in insects. However, the genome-wide distribution of chitinase-like gene family in *Bemsia tabaci*, a destructive pest worldwide, is still elusive. With the help of bioinformatics, we annotated 14 genes that encode putative chitinase-like proteins, including ten chitinases (Cht), three imaginal disk growth factors (*IDGF*), and one endo-β-N-acetylglucosaminidase (*ENGase*) in the genome of the whitefly, *B. tabaci*. These genes were phylogenetically grouped into eight clades, among which 13 genes were classified in the glycoside hydrolase family 18 groups and one in the *ENGase* group. Afterwards, developmental expression analysis suggested that *BtCht10*, *BtCht5,* and *BtCht7* were highly expressed in nymphal stages and exhibit similar expression patterns, implying their underlying role in nymph ecdysis. Notably, nymphs exhibited a lower rate of survival when challenged by dsRNA targeting these three genes via a nanomaterial-promoted RNAi method. In addition, silencing of *BtCht10* significantly resulted in a longer duration of development compared to control nymphs. These results indicate a key role of *BtCht10*, *BtCht5,* and *BtCht7* in *B. tabaci* nymph molting. Our research depicts the differences of chitinase-like family genes in structure and function and identified potential targets for RNAi-based whitefly management.

## 1. Introduction

Chitin, the structurally simplest glycosaminoglycans, which is a β-1,4-linked linear homopolymer of N-acetylglucosamine, distributes widely in arthropods and several microbes [1]. In insects, as an insoluble polysaccharide, chitin is a momentous structural component of exoskeleton, trachea and the peritrophic matrix (PM) as well as salivary gland, eggshells, and muscle attachment points [2]. Due to the rigidity and chemical steadiness of chitin, insects build a rigid exoskeleton to protect themselves from environment injuries and pliable PM to keep their midgut safe from mechanical damage and pathogen infections. However, chitin is also the key factor that constrains the growth and development of insects and in order to grow, insects have to periodically shed off their old exoskeleton and turn over PM, and then the new version are rebuilt for defending themselves [3,4].

Chitinase (EC 3.2.1.14, endochitinase) is a kind of glycosyl hydrolases that hydrolyze the β-1,4-glycosidic linkages in chitin. In addition, they extensively exist in nature as they are found in species across all kingdoms, while they function very differently and are mainly involved in digestion, arthropod molting, defense/immunity, and pathogenicity [5]. Insect chitinase is divided into Glycoside hydrolase family 18 (GH18; PFAM database accession: PF00704, http://pfam.xfam.org/ accessed date, 14 March 2021) chitinase-like superfamily which include chitinases with catalytic activity as well as those lacking chitinase activity, such as imaginal disk growth factors (IDGFs), endo-β-N-acetylglucosaminidases (ENGases), stabilin-1 interacting chitinase-like proteins (SI-CLPs) and chitolectins [6]. The proton donor glutamic acid in the active site of chitinase is important for the hydrolysis of β-1,4-glycosidic bond in chitin and its substitution in the enzymatically inactive GH18 members accounts for the lack of chitinolytic activity, even if they may still do well in binding chitin with high infinity [6,7]. Chitinase is critical in helping insects shed off old cuticle and PM turnover. During molting, insects use chitinases to hydrolyze the structural polysaccharide in their exoskeletons and gut linings and digested the insoluble polymeric chitin into soluble, yielding low molecular mass multimers of N-acetyl-beta-D-glucosamine (GlcNAc), such as chitotetraose, chitotriose, and chitobiose [1,8,9,10].

Numerous chitinases and chitinase-like proteins in different orders of insects have been annotated and characterized with the help of functional genomics, which indicates that there are rather large and diverse groups of genes encoding chitinase-like proteins. Based on the difference in primary structures and domain architectures, insect chitinases have been classified into eight phylogenetic groups (group I–VIII) which are commonly adopted [2,5] and two additionally identified groups, group IX and X, makes it into ten groups [11]. The systematic analysis of biological functions of chitinases and chitinase-like proteins in *Tribolium castaneum* suggested that chitinase genes in group I and II are engaged in insect exuviate and egg hatching, chitinase genes in group III mainly involved in regulating abdominal contraction and wing expansion, while the imaginal disk growth factor genes in group V have little relation to cell proliferation or differentiation of the imaginal disk but needful for adult emergence [12]. Chitinases and chitinase-like proteins in *Nilaparvata lugens*, one of the typical hemimetabolous insects, were also studied for function analysis by RNA interference experiments. Five genes were found to be involved in *N. lugens* moulting and resulted in lethal phenotypes after RNAi-treated. Similar to *T. castaneum*, five genes related to moulting in *N. lugens* are either from group I, group II, or group III, except one gene which was tentatively placed in group IV [13]. In *Acyrthosiphon pisum*, there were also several chitinases which could not be clustered [4]. More chitinase sequences need to be analyzed, as there are still chitinases in these hemimetabolous insects that could not be clustered into a certain group. From this point of view, the inadequate annotation data on chitinase-like genes of different insect species may account for the inability to classify certain genes. Hence the genome-wide identification of chitinase-like gene family in more insect species is necessary for accelerating the classification of insect chitinases.

The intake of double-stranded RNA (dsRNA) by direct injection or oral feeding leads to RNA interference (RNAi) in insects, which provides an efficient tool for gene function analysis [13,14,15,16]. In this study, we applied a nanomaterial-promoted RNAi method which previously reported to improve dsRNA penetration through body wall and RNAi-induced mortality in *Aphis glycines* [17]. It is always necessary to have concern for the safety of non-targets when RNAi-based method of controlling pests is to be applied in the field. As we know, dsRNA can specifically target gene transcript without affecting non-target species, which on the one hand ensures the safety of non-targets. On the other hand, research showed that silencing of the aphid hemocytin gene (*Hem*) by a nanocarrier-promoted RNAi method dramatically decreased aphid population density, while the cell survival ratios of S2 cells with the same treatment was more than 96%, which demonstrates the safety of the RNAi method to non-targets [17]. Such transdermal delivery system is convenient to operate and of high efficacy as well as safe to non-targets, so it may provide a promising tool to perform RNAi in the laboratory or field [17,18,19,20].

As one of the hemimetabolous insects, *Bemisia tabaci* (Gennadius) (Hemiptera: Aleyrodidae) is a notorious pest which causes massive damage to crops in temperate climate around the world [21]. It uses specialized stylet to digest nutrients from plant phloem and transmits numerous plant virus causing severe diseases [22,23]. Research indicates that *B. tabaci* is a species complex comprising at least 24 cryptic species, among which Mediterranean (MED) and Middle East-Asia Minor 1 (MEAM1) are the most invasive and destructive [21,24,25]. MED and MEAM1 are morphologically indistinguishable and differ in host preference, virus transmission, and insecticide resistance [26,27,28]. Besides, MED is recognized as more resistant and able to develop stronger resistance to insecticides than MEAM1 [28,29], which is assumed to be a main reason for the displacement of MEAM1 by MED in many areas. Particularly in China, MED has displaced MEAM1 since 2005, apparently due to the frequent use of insecticides over past decades [30,31]. More severely, despite chemical control with insecticides remaining effective in controlling *B. tabaci*, the prevalent use of insecticides has resulted in emerging resistance to many classes of insecticides worldwide, even causing a serious threat to agricultural system, food safety and public health [32,33,34,35,36,37,38,39,40,41]. To maintain sustainable pest management, therefore, new ideas for *B. tabaci* control need to be developed.

Chitinases play an important role in regulating insect growth and development, which might be a promising target for insecticide-based control of *B. tabaci* [8]. Although several genes in MEAM1 genome have been annotated as chitinases [42]; however, genome-wide characterization of chitinase-like genes in MED *B. tabaci* remains unknown to date. In this study, we searched the MED *B. tabaci* genome [43] to identify genes encoding chitinases and chitinase-like proteins and annotated their cDNA sequences. The gene architectures, phylogenetic relationships, expression patterns and primary biology function analysis of chitinases were analyzed to elucidate chitin metabolism in whitefly growth and development.

## 2. Materials and Methods

### 2.1. Insect Rearing and Sample Preparing

*Bemisia tabaci* MED population was maintained on cotton plant (Gossypium herbaceum L. cv. Zhongmian 49) at 25 ± 1 °C with a photoperiod of 16 h light: 8 h darkness and 60 ± 10% relative humidity. A mitochondrial cytochrome oxidase I (mtCOI) marker was used to ensure the purity of the strain every 3~5 generations [44].

Different developmental stages (egg, 1st and 2nd instar nymph, 3rd instar nymph, 4th instar nymph and newly emerged adult) of *B. tabaci* MED were collected, then rapidly frozen in liquid nitrogen and stored at −80 °C, which were subsequently as samples to analyze the expression pattern of the 14 chitinase-like genes.

### 2.2. Identification of Chitinase and Chitinase-Like Genes in B. tabaci

Amino acid sequences of chitinase and chitinase-like genes from *N. lugens* (Hemiptera), *A. pisum* (Hemiptera), *Drosophila melanogaster* (Diptera), *T. castaneum* (Coleoptera), *Apis mellifera* (Hymenoptera), *Manduca sexta* (Lepidoptera), and *Pediculus humanus corporis* (Anoplura) were downloaded from National Center for Biotechnology Information (NCBI) (https://www.ncbi.nlm.nih.gov/ accessed date, 14 March 2021) and Ensembl Metazoa databases (http://metazoa.ensembl.org/index.html/ accessed date, 14 March 2021) as queries to search against *B. tabaci* genome and transcriptome [43,45] for screening chitinase and chitinase-like genes in *B. tabaci*, using the BLASTX algorithm with an E-value threshold of 10^−5^. The candidate *B. tabaci* chitinase-like family genes were confirmed by searching the BLASTX algorithm against the non-redundant protein sequence (NR) database of NCBI. Putative chitinase and chitinase-like genes were manually rectified by comparison with assembled expressed sequence tags (ESTs) [45].

### 2.3. RNA Isolation, cDNA Synthesis and Full-Length cDNA Cloning

Extraction of total RNA was carried out by using TRIzol reagent following the manufacturer’s instructions (Invitrogen, Carlsbad, CA, USA). NanoDrop 2000 (Thermo Fisher Scientific Inc., Waltham, MA, USA) was applied to quantify the concentration of RNA extracted, and purity was evaluated on 1% agarose gels. For qRT-PCR analysis, total RNA was reverse-transcribed into cDNA with a PrimeScript RT kit (Perfect Real Time) (TaKaRa, Dalian, China) and a PrimeScript™ II 1st strand cDNA synthesis kit (TaKaRa, Dalian, China) was also used to synthesize 1st strand cDNA for gene cloning. The cDNA was stored at −80 °C for later use. Based on genome and transcriptome [43,45], DNA sequences of the 14 *B. tabaci* chitinase-like proteins were obtained and then used as templates to design gene-specific primers with PRIMER PREMIER 5 program [46]. These primers were used to amplify cDNAs of the chitinase-like genes and to obtain the complete open reading frames (ORFs) (Appendix A). The PCR conditions were confirmed empirically for the cloning of each chitinase-like genes (Appendix A). Then the PCR products were used for electrophoresis, and the DNA binds of expected size were excised from the agarose gel, after which gel purification was carried out by using a DNA gel extraction kit (TransGen, Beijing, China). These purified PCR products were introduced into the pEASY-T1 vector (TransGen) and four independent clones were sequenced for each cDNA.

### 2.4. Classification of B. tabaci Chitinases and Chitinase-Like Proteins by Construction of Phylogenetic Trees

For analysis of evolutionary relationships, deduced amino acid sequences of chitinase-like proteins selected from seven representative insect species in six different orders were aligned with chitinase-like proteins in *B. tabaci* by CLUSTALX and then phylogenetic trees were constructed using the MEGA7 program with ML method [47]. The bootstrap support of tree branches was evaluated by resampling amino acid positions 1000 times.

### 2.5. Sequence Analysis of Exon-Intron Distribution and Domain Structure

Exon-intron distribution of the 14 genes in *B. tabaci* genome was analyzed based on alignments of putative open reading frames against their corresponding genomic sequences and CLUSTALX program was applied for multiple sequence alignments [48]. Amino acid sequences of the 14 genes were respectively subjected to the Simple Molecular Architectural Research Tool (SMART; http://smart.emblheidelberg.de/ accessed date, 14 March 2021) and another BLAST tool on the NCBI website (https://www.ncbi.nlm.nih.gov/Structure/cdd/wrpsb.cgi/ accessed date, 14 March 2021) for conserved domains finding.

### 2.6. Expression Pattern Analysis of Chitinase and Chitinase-Like Genes in B. tabaci by Real-Time qPCR (qRT-PCR)

Expression levels of chitinase-like genes in *B. tabaci* were determined by qRT-PCR with gene-specific primers designed by Primer premier 5.0 (Appendix A). ABI PRISM 7500 Real-time PCR System (Applied Biosystems, Foster City, CA, USA) was used for conduction of qRT-PCR with a 20-μL reaction system containing 0.4 μL of 50 × ROX reference dye (TIANGEN, Beijing, China), 0.6 μL of each specific primer, 1 μL of cDNA template, 7.4 μL of ddH2O, and 10 μL of 2 × SuperReal PreMix Plus (SYBR Green) (TIANGEN, Beijing, China). The qRT-PCR program was as follows: 95 °C for 10 min (initial denaturation), followed by 40 cycles of 95 °C for 5 s (denaturation), 60 °C for 15 s (annealing), and 72 °C for 35 s (elongation). qRT-PCR primers which meet the amplification efficiencies of 90%–110% were used and listed in Appendix A. Relative expression levels were quantified using the 2^−ΔΔCt^ method [49]. Two reference genes 60S ribosomal protein L29 (RPL29) (GenBank accession no. EE596314) and elongation factor 1 alpha (EF1-α) (GenBank accession no. EE600682) were used for normalization of target genes expression [50]. For each sample, three biological replicates and four technical replicates were performed.

### 2.7. dsRNA Synthesis and RNA Interference (RNAi) on BtCht5, BtCht7 and BtCht10

Gene-specific primers with a T7 promoter, used for amplification of target gene fragments, were designed by a web-based dsRNA design tool (https://www.flyrnai.org/cgi-bin/RNAi_find_primers.pl/ accessed date, 14 March 2021). The T7 Ribomax™ Express RNAi System (Promega, Madison, WI, USA) was used for dsRNA synthesis according to the manufacturer′s instructions. The quality and integrity of dsRNA was ensured by gel electrophoresis and quantification was carried out by using a Nanodrop spectrophotometer. The primers used are listed in Appendix A. The dsRNA was stored at −80 °C until use.

Since *BtCht5*, *BtCht10,* and *BtCht7* respectively belongs to Group I, Group II; and Group III, in which genes were reported to engage in old cuticle degradation and expression patterns revealed their high level of expression in nymphal stage, we selected these genes for function analysis via a nanomaterial-promoted RNAi method [17,51].

Cotton plants with two true leaves were placed in plastic cups and plant leaves were determined to be free of any eggs or nymphs using a microscope before infestation with *B. tabaci*. To obtain synchronized whitefly nymphs, approximately 400 MED adults were placed on the leaves to lay eggs for 4 h and after that adult whiteflies were removed.

Whitefly nymph in second instar stage was selected for gene function analysis and ten days of recording were generated to compare the difference of gene-silenced nymphs with control nymphs in survival rate and the developmental duration from second instar to third instar. Once the second instar nymph emerged, the dsRNA droplet was dripped on the nymph with a 10-μL pipette. The concentration of the dsRNA solution was 0.5 μg/μL, as the solution contained 1 μg/μL dsRNA and an equal volume of nanomaterial provided by Professor Jie Sheng from China Agricultural University. Cotton plants with dsRNA treated nymphs were placed in an illumination incubator which ambient temperature was set at 25 °C with a photoperiod of 16 h light: 8 h darkness and 60% relative humidity. The RNAi efficiency was determined after 48 h. Each treated nymph was circled with a fine tip non-toxic Sharpie^®^ marker and survival was recorded daily to generate a survival curve for each treatment. The enhanced green fluorescent protein (EGFP) gene was used as a control. Lethal phenotypes of gene-silenced *B. tabaci* nymphs were captured by a stereomicroscope (Leica, M205C, Solms, Germany).

### 2.8. Statistical Analysis

The results of the survival bioassays were subjected to survival analysis performed using the Kaplan–Meier estimators (Log-rank method) with GraphPad Prism 8 [52,53]. Log-rank (Mantel-Cox) test was used to calculate the curve comparison. For analysis of developmental duration, unpaired T test [54] was used. The values were calculated as the mean values and standard errors of the means.

## 3. Results

### 3.1. In Silico Identification and Classification of Chitinase and Chitinase-Like Genes in B. tabaci

Amino acid sequences of seven insect species were used as queries to do BLAST searches against the genomic and transcriptomic databases of *B. tabaci* [43,45]. BLAST searches identified 14 chitinase-like genes including 13 chitinase genes and one ENGase gene.

To further classify the 14 putative chitinase-like genes in B. tabaci, phylogenetic analysis was performed. Deduced amino acid sequences of chitinase-like genes in B. tabaci and seven other insect species from representative orders were aligned and analyzed. Protein sequences were then subjected to phylogenetic analysis using the maximum likelihood method (Jones-Taylor-Thornton model; Nearest-Neighbor-Interchange ML heuristic method) with MEGA 7 software (Figure 1). These chitinase-like genes from eight herbivore species were clustered into 12 individual groups (GH18 groups I–X, Lepidoptera-specific chitinase h group and group ENGase). The 14 chitinase-like genes in *B. tabaci* were clustered into ten groups (group I–VIII, X and ENGase), but none of these genes were classified into group IX. It is like another two pierce-sucking insects (*N. lugens* and *A. pisum*) that group IX in *B. tabaci* was absent. The most discrepant group among all groups should be group IV, among which sequences that could not be clustered into any other groups were provisionally classified into group Ⅳ. Three *B. tabaci* chitinase-like genes (*BtCht4*, *BtCht8* and *BtCht9*) were also tentatively placed in group IV. Except that group V has three genes (*BtIDGF1-3*), the remaining eight groups each has one single gene: *BtCht5* in group I, *BtCht10* in group II, *BtCht7* in group III, *BtCht6* in group VI, *BtCht2* in group VII, *BtCht11* in group VIII, *BtCht3* in group X and BtENGase in group ENGase.

### 3.2. Characterization of the Domain Structures and Exon-Intron Distribution for B. tabaci Chitinase-Like Proteins

Domain structures of the 14 chitinase-like proteins were predicted by using their inferred amino acid sequences (Figure 2). Except for BtENGase, the rest of the 13 chitinase-like proteins in *B. tabaci* all possessed the GH18 (PFAM database accession: PF00704) chitinase-like superfamily domains. Among these 13 proteins, *BtCht7* and *BtCht10* respectively had two and three copies of GH18 domain, while the remaining ones each had a single copy. Additionally, BtENGase had a GH85 (PFAM database accession: PF03644) ENGase domain rather than a GH18 domain.

As previous studies found, there are four conserved regions in the protein sequence of insect chitinases [8,10,55]. Conserved region I (CR_I) is referred to the amino acid sequence of KxxxxxGGW, where x is an undetermined amino acid. Conserved region II (CR_II) stands for the sequence of FDGxDLDWEYP, which is considered to play an important role in the enzyme activity of chitinase and the E residue is a potential proton donor in the catalytic mechanism. Conserved region III and IV (CR_III and CR_IV) are represented by the sequence of MxYDxxG and GxxxWxxDxD, respectively. The four conserved regions in the GH18 domain were also found in the *B. tabaci* chitinase-like proteins (Figure 3). All the four regions were well conserved in *BtCht5*, *BtCht10* and *BtCht7*, and these three genes were separately clustered into group I, group II and group III, where genes were reported to be involved in insect molting, egg hatching, abdominal contraction, and wing expansion [12]. These four regions in *BtCht2*, *BtCht4*, *BtCht8*, *BtIDGF2,* and *BtCht11* were moderately conserved and they may all have a chitinase activity because of the conservation of the key glutamate in CR_II. In *BtIDGF3* and *BtIDGF1*, the key active site glutamate was replaced with glutamine and lysine in CR_II, suggesting that they may lack chitinase activity. Nonetheless, all the four regions in *BtCht3* were poorly conserved.

All of the 14 chitinase-like genes in *B. tabaci* were matched to a certain scaffold sequence in *B. tabaci* genome and the exon-intron distribution were shown (Figure 4). These genes were highly discrepant in both gene sizes and the number of exons and introns. *BtCht8* and *BtCht9* both had only one exon and no introns were present, while *BtCht10* had the most exons reaching up to 31. Besides, the sizes of their exons varied a lot, ranging from 0.4 kb to 4 kb and sizes of introns also differed greatly with a distribution from 0.5 kb to 16 kb.

### 3.3. Developmental Expression Patterns of Chitinase-Like Genes in B. tabaci

Although the majority of the chitinase-like genes expressed in various immature stages of *B. tabaci*, *BtCht3*, *BtCht4*, *BtIDGF1-3*, and *BtCht2* were rarely expressed in the egg stage, suggesting these genes may not be implicated in egg hatching or development. *BtCht5*, *BtCht7,* and *BtCht10* had similar expression patterns, and as their expressions were much higher in nymph than that of adult. Potential roles of these genes in immature molting are implied but to be verified. Interestingly, for *BtIDGF1-3* and *BtCht2*, the transcript levels were peaked in adult stage, may suggest that these genes may be engaged in adult growth and development (Figure 5).

### 3.4. Phenotypes and RNAi Effects of Insects Treated with Double-Stranded RNA (dsRNA) for Chitinase-Like Genes BtCht5, BtCht10 and BtCht7 in B. tabaci

Given the high expression levels of *BtCht5*, *BtCht10,* and *BtCht7* in nymph, and that previous studies support that they may have an important role in conferring juvenile molting, these chitinase-like genes were selected in the RNAi studies and subsequent phenotype observations. The application of ds*BtCht10*-RNA, ds*BtCht5*-RNA, and ds*BtCht7*-RNA reduced the transcript levels of *B. tabaci* by 49% (t = 2.810; df = 4; *p* = 0.0483), 70% (t = 3.745; df = 4; *p* = 0.02) and 57% (t = 10.47; df = 4; *p* = 0.0005), respectively, at 48 h after dsRNA treatment (Figure 6A). Among all the second instar nymphs, 83% of ds*EGFP*-treated nymphs, 49% of ds*BtCht10*-treated nymphs, 52% of ds*BtCht5*-treated nymphs, and 49% of ds*BtCht7*-treated nymphs successfully shed their old cuticles and developed into third instar nymphs (Figure 6B). As survival curves indicated, after ten days, the survival rates of nymphs respectively treated with ds*BtCht10*-RNA, ds*BtCht5*-RNA, and ds*BtCht7*-RNA were significantly decreased to 38% (x^2^ = 19.28; df = 1; *p* < 0.0001), 32% (x^2^ = 29.24; df = 1; *p* < 0.0001) and 28% (x^2^ = 34.72; df = 1; *p* < 0.0001), while that of ds*EGFP*-treated nymphs was 74% (Figure 6C). Interestingly, the developmental duration of nymphs treated with droplets of ds*BtCht10*-RNA was 5.4 days, which was markedly postponed by 1.5 days when compared to control nymphs treated with ds*EGFP* (t = 5.318; df = 103; *p* < 0.0001) (Figure 6(D1)). However, silencing of *BtCht5* or *BtCht7* did not exhibit similar RNAi effects (Figure 6(D1,D2)). Lethal phenotypes of silenced nymphs were also observed and captured. Control nymphs were plump and oval-shaped as well as saturated with hemolymph and looked dynamic, while gene-silenced nymphs were shrunken and shriveled as well as mummified (Figure 7).

## 4. Discussion

In nature, chitinase is comprehensively distributed and serves as a generalist which engages in digestion, arthropod molting, defense/immunity, and pathogenicity [5]. Besides, as for insects, chitinases are essential for growth and development across their lifetime. Therefore, understanding of insect chitinase biology is of quite importance for providing promising targets of pest control. In this study, we identified 14 chitinase-like genes in a disastrous agriculture pest *B. tabaci* for the first time and functionally demonstrated that some chitinases may play a vital role in *B. tabaci* juvenile exuviation, which armed our arsenal to cope with such intractable pest by interfering the expression of chitinases.

The increasing availability of genomic data from different insect species have greatly accelerated the genome-wide annotation of various gene families, which also helped the identification of chitinase-like genes in *B. tabaci*. By searching and screening of the *B. tabaci* genome, 14 genes encoding chitinases and chitinase-like proteins were first identified. It was consistent with the two hemimetabolous sap-feeding insects (*N. lugens* and *A. pisum*) that *B. tabaci* had a relatively small number of chitinase-like proteins when compared to other insect species, especially those in Diptera and Coleoptera [4,10,13]. It may be explained that hemimetabolous insects, such as *B. tabaci*, undergo an incomplete metamorphosis which is much milder and fewer enzymes and less energy are needed to help reconstruct the new integument [4]. In addition, this highly simplified morphology and auxanology of whitefly nymphs is in tune with their sessile feeding habit [56].

Interestingly, among the three hemimetabolous sap-feeding species, *B. tabaci* exhibited the most chitinases, which was two more than *N. lugens* and five more than *A. pisum*. Molecular phylogenetic analyses revealed that there were three *B. tabaci* IDGF genes (*BtIDGF1-3*) which were clustered in Group V, wheras *N. lugens* and *A. pisum* had only one IDGF separately. Besides, three genes (*BtCht4*, *BtCht8* and *BtCht9*) in *B. tabaci* were failed to be clustered and tentatively placed in group IV; however, for *N. lugens* and *A. pisum*, there was only one gene respectively. These discriminations may account for the difference in gene quantity of chitinases between *B. tabaci* and the other two hemimetabolous insects. It was worth mentioning that *NlCht3* and *ApCht8* were previously placed in group IV while now could be clustered with *BtCht3* and classified as group X genes (Figure 1). *BtCht3* has an N-terminal signal peptide ahead of the GH18 catalytic domain followed by two very closely spaced tandem chitin-binding domains and a very long C-terminal stretch ending with a third CBD (Figure 3), which is in consistence with previous studies of genes in group X [11]. However, *BtCht4*, *BtCht8,* and *BtCht9* along with *ApCht7* were still tentatively divided into Group IV because they could not be clustered into other groups. In this case, with increasing numbers of chitinase-like genes annotated in different insect species, it is likely that members in group IV probably would be divided into some other brand-new groups, and the potential functions of these genes should be noticed and studied.

In this study, we explored gene expression patterns of each chitinase and chitinase-like genes in *B. tabaci* via qRT-PCR. Results revealed that six genes were highly expressed in adult stage and most of them do not have chitinase catalytic activity since the key motif of CR_Ⅱ is mutated (Figure 3 and Figure 5). Among these genes, there are three *BtIDGF*s and they all shared a similar expression pattern. IDGFs were reported to have multiple functions in insect and mammalian cells, involving in regulation of cell proliferation and acting as chitolectins which interact with cell surface receptors [57,58]. The *D. melanogaster IDGF2* was turned out to be a trophic factor involved in energy balance, detoxification, and innate immunity, which promoted cellular and organismal survival of *D. melanogaster* [59]. *IDGF4* in *Bactrocera dorsalis* played an essential role in regulating its development and temperature adaptation [60]. Although functionality of IDGFs in *B. tabaci* has yet to be determined, it is implied by this research that IDGFs played essential roles in adult performance. In addition, it might be inferred that *BtIDGF*s also have some important roles in detoxification of xenobiotics, which associates with the severe resistance of *B. tabaci* to kinds of insecticides [29,61,62].

Egg and Early nymphal stages of whitefly were considered less detrimental to crops since their limited feeding capacity [29]. In addition, it is sensible and rational to disturb the normal development of whitefly in their early nymphal stage to hold back further loss. Therefore, three chitinase-like genes respectively from Group I, Group II and Group III, which were highly expressed in nymphal stages, were selected as targets for analysis of their potential roles in nymphal development. As results showed, silencing of *BtCht10*, *BtCht5* and *BtCht7* led to a significant higher rate of death over ten days of recoding and fewer nymphs survived a molting course in the gene-silenced groups than control (Figure 6). These results were in consistence with previous studies, which clarified that chitinase-like genes from Group I, Group II, and Group III were of key importance in old cuticle degradation [1,12,13]. These results along with previous research, enriches our knowledge about how insect chitinases are involved in chitin synthesis and degradation so that certain enzymatic steps can be perturbed and potentially act as targets of pest control [63]. Consequently, *BtCht10*, *BtCht5,* and *BtCht7* are as key genes involved in *B. tabaci* chitin degradation, which provides a promising idea of *B. tabaci* management by interfering the expression of these representative genes.

## 5. Conclusions

In summary, 14 chitinase-like genes were identified in *B. tabaci* genome and these genes were divided into ten distinct groups through phylogenetic analysis, by which there were no *B. tabaci* chitinase-like genes clustered in group IX. Then, pattern expression analysis showed that these genes displayed rather different models in disparate developmental stages, which may be associated with their discrepant biological functions. Additionally, three genes with high transcript levels in nymph stage turned out to play a key role in *B. tabaci* nymph molting by a nanomaterial-promoted RNAi method. Our data could clarify the structures, phylogeny, categories, expression models, and biological functions of *B. tabaci* chitinase-like family genes, and provide further understanding on insect chitinases, which may advance novel insect pest management strategies.

## Figures and Tables

**Figure 1 insects-12-00254-f001:**
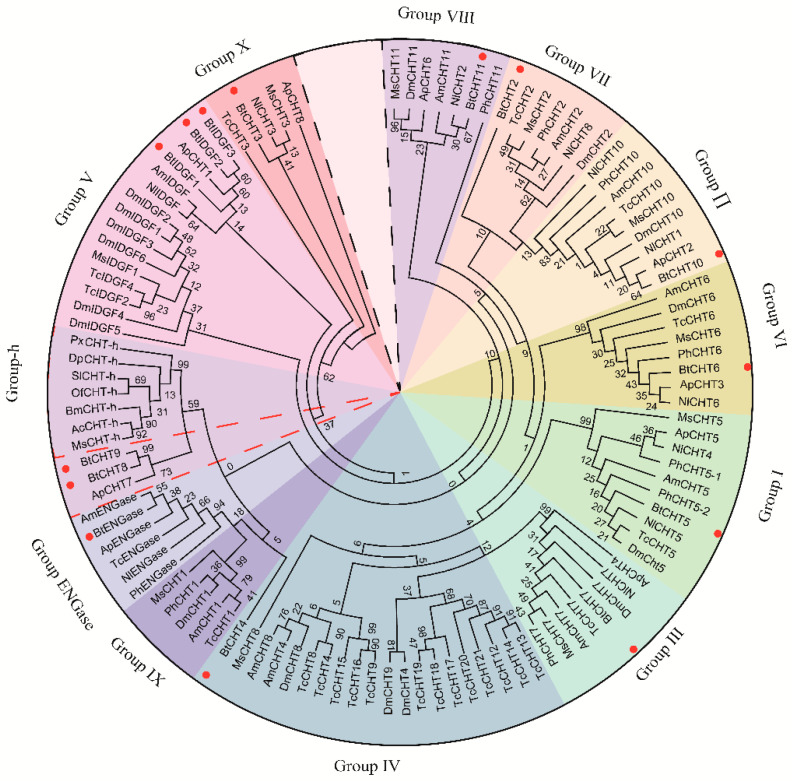
Phylogenetic tree of chitinase-like proteins from eight insect species. MEGA7 software was used to generate the phylogenetic tree with the maximum likelihood method. A bootstrap analysis of 1000 replicates was applied, and bootstrap values are shown in the cladograms. Dm stands for *D. melanogaster*, Bm for *B. mori*, Ph for *P. humanus*, Nl for *N. lugens*, Ap for *A. pisum*, Tc for *T. castaneum*, Am for *A. mellifera* and Bt for *B. tabaci*. Ac *for Agrius convolvuli*, Ms for *Manduca sexta*, Of for *Ostrinia furnacalis*, Px for *Papilio xuthus*, Sl for *Spodoptera litura*. Chitinase-like proteins from *B. tabaci* are marked in red dots.

**Figure 2 insects-12-00254-f002:**
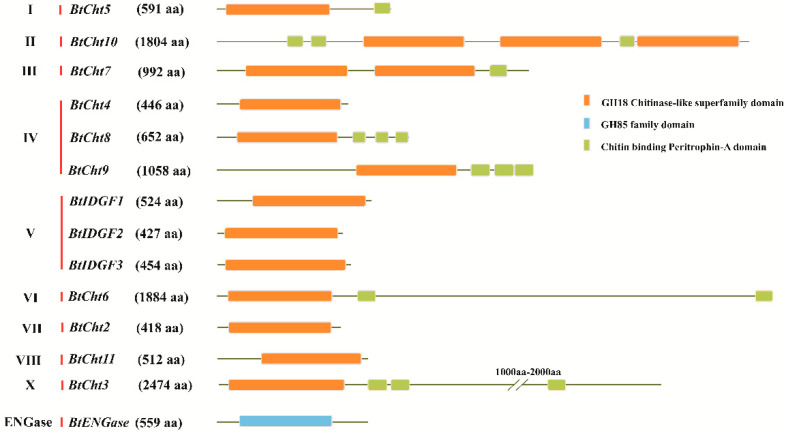
Conserved domain architectures of 14 chitinase-like genes in *Bemisia tabaci*. Based on chitinase-like gene sequences, the deduced amino acid sequences were obtained. These deduced protein sequences of *B. tabaci* chitinase (*BtCht*) genes were then subjected to SMART software and a conserved domain- searching tool (National Center for Biotechnology Information) for domain architectures predicting. Orange boxes indicates the GH18 chitinase-like superfamily domain, boxes in green stand for Chitin-binding Peritrophin-A domain and GH85 family domain is implied by a light blue box. ENGase, endo-β-N-acetylglucosaminidase; GH, glycoside hydrolase; IDGF, imaginal disk growth factor.

**Figure 3 insects-12-00254-f003:**
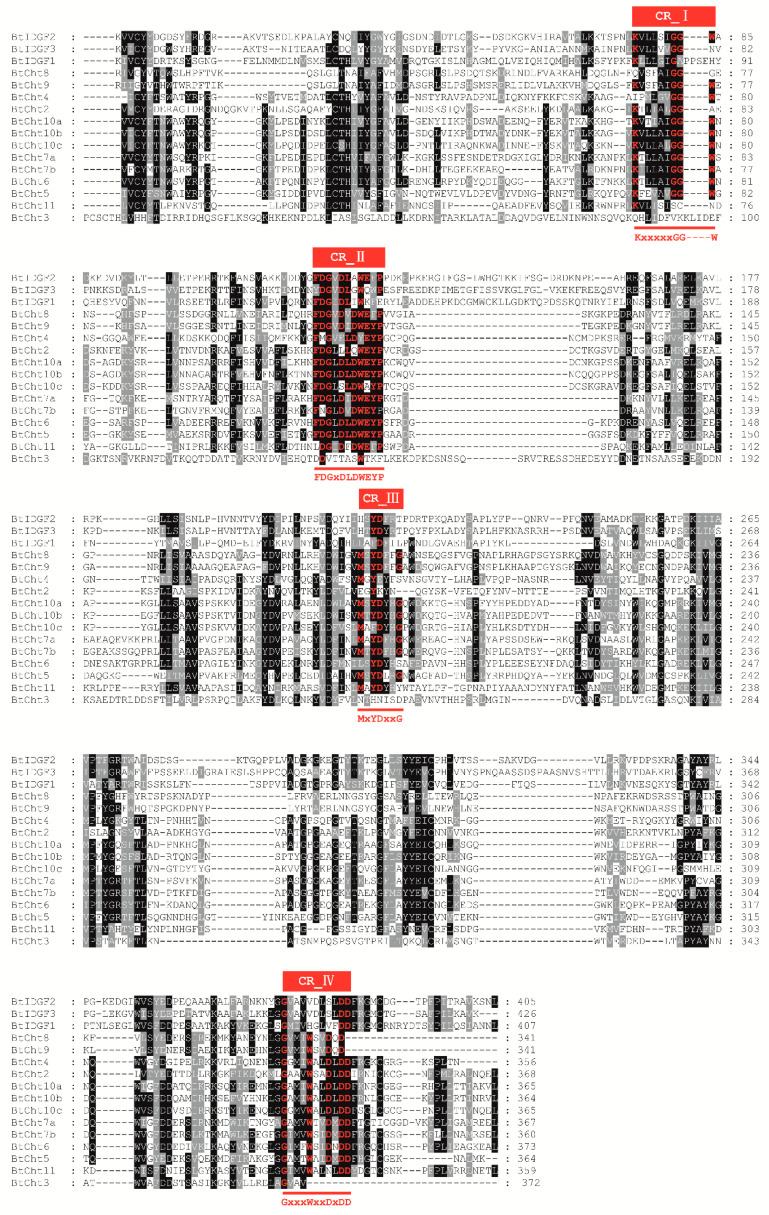
Conserved regions in the glycoside hydrolase family 18 (GH18) domain of 13 chitinase and chitinase-like proteins in *B. tabaci*. CLUSTALX software was used to conduct the alignment of amino acid sequences of the catalytic domains in GH18 family enzymes. It highlights with different levels of gray and black shading where residues are the same as the consensus of residues for the column. Black shading indicates that all residues are the same in the column. Different sequence homologies are implied by different shading. Regions underlined are the four conserved motifs represented by the sequences KxxxxxGGW, FDGxDLDWEYP, MxYDxxG and GxxxWxxDxDD. Highly conserved residues are marked in red. CR, conserved region.

**Figure 4 insects-12-00254-f004:**
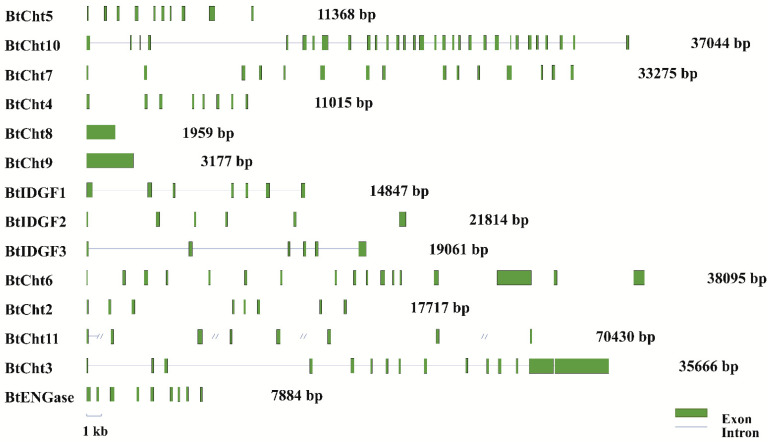
Exon−intron distributions of 14 *B. tabaci* chitinase-like genes. Genomic sequences and putative cDNA sequences were compared to determine the exon-intron distribution of each chitinase-like genes. Exons are represented by boxes in green and lines separating these green boxes are on behalf of the introns. Cht, chitinase; ENGase, endo-β-N-acetylglucosaminidase; IDGF, imaginal disk growth factor.

**Figure 5 insects-12-00254-f005:**
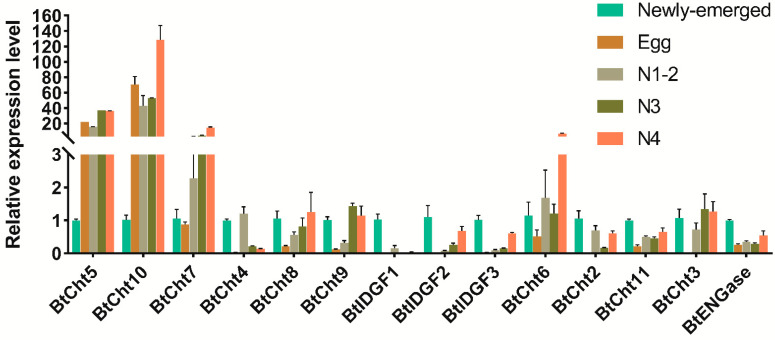
Expression patterns of 14 chitinase-like genes in different development stages of *B. tabaci* by quantitative real-time PCR (qRT-PCR). Total RNA was extracted from samples including mixture of first and second instar nymphs (N1-2), third instar nymphs (N3), forth instar nymphs (N4) and newly emerged adults. The *B. tabaci* elongation factor 1 alpha (EF1-α) and 60S ribosomal protein L29 (RPL29) were used as an internal control. The real-time qPCR results were analyzed by the △△Ct (Cycle threshold) method. Three biological replicates were performed for each gene based on independent RNA sample preparations. Cht, chitinase; ENGase, endo-β-N-acetylglucosaminidase; IDGF, imaginal disk growth factor.

**Figure 6 insects-12-00254-f006:**
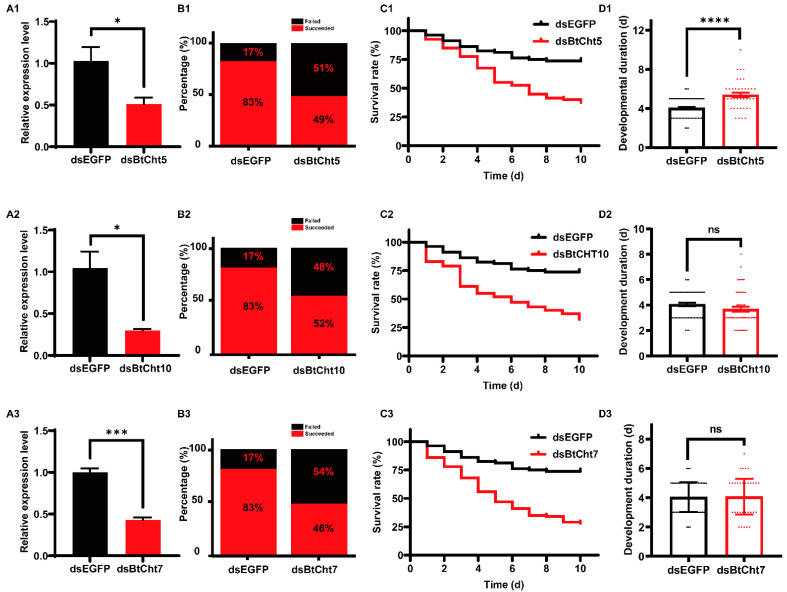
RNAi effects of three *B. tabaci* chitinases (*BtCht10*, *BtCht5* and *BtCht7*) on survival and developmental duration from second instar nymph to third instar nymph. (**A**) Efficiency of RNA interference on target genes. Nymphs were collected after two days of treatment with dsRNA and expression levels of target genes were quantified by qRT-PCR. Error bars indicate standard error of mean (*n* = 3). (**B**) Percentage of nymphs that succeeded (indicated by red) or failed (indicated by black) to survive a molt course after dsRNA (0.5 µg/µL) treatment. (**C**) Survival curves of *B. tabaci* after exposure to dsRNA (0.5 µg/µL). For each treatment about 80~100 s instar nymphs were continuously recorded for 10 days and data were used to make the survival curves. (**D**) Developmental duration (from second instar to third instar) of nymphs treated with dsRNA (0.5 µg/µL). All second instar nymphs that successfully experienced a molt were used to analyze the developmental duration. * *p* < 0.05, *** *p* < 0.001, **** *p* < 0.0001.

**Figure 7 insects-12-00254-f007:**
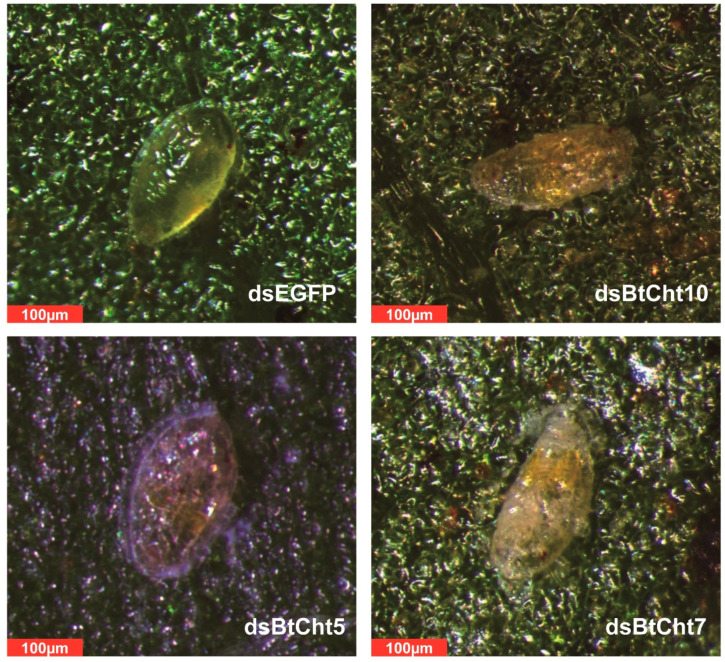
Lethal phenotypes of nymphs treated with dsRNA for *B. tabaci* chitinases (BtChts). dsRNA targeting gene of enhanced green fluorescent protein (EGFP) was used as negative control. Nymphs treated with ds*BtCht10*, ds*BtCht5* and ds*BtCht7* exhibited similar phenotypes and high mortality.

## Data Availability

Detailed data are available upon request.

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
