# Peer review of "Genome-Wide Identification and Analysis of Chitinase-Like Gene Family in Bemisia tabaci (Hemiptera: Aleyrodidae)"

_insects, 2021, doi:10.3390/insects12030254_

Round 1

Reviewer 1 Report

In the present study, 14 chitinase-like genes were identified in B. tabaci genome, and nanomaterial-promoted RNAi proved that BtCht10, BtCht5 and BtCht7 resulted in significant increase of death rate on B. tabaci nymphs, and the developmental duration was noticeably postponed for BtCht2-silenced nymphs. The results may advance novel insect pest management strategies. The minors as followings:

P198: Please give more information and the operation procedure about “nanomaterial-promoted RNAi method”?

P203: Why collect Whitefly nymph in second-instar stage not other stages to make gene function analysis.

P248: Fig1 May be it will be better to list the sequences numbers of these target species.

Author Response

Dear reviewer:

Thanks for your patient revision and professional advice on our article. We have made point-by-point modifications based on the valuable comments you gave us and details are as follow:

Point 1: P198: Please give more information and the operation procedure about “nanomaterial-promoted RNAi method”?

Response 1: Thanks for your suggestion. We have referenced the original papers about the nanomaterial we used in this study and have made some detail descriptions about the nanomaterial-promoted RNAi method in line 233-236.

Point 2: P203: Why collect Whitefly nymph in second-instar stage not other stages to make gene function analysis.

Response 2: Thanks for your question. Development rates between whitefly nymph individuals are usually discrepant and such difference gets more significant when it develops into third or fourth-instar stage. Hence, the earlier stage the better it is for development duration analysis. While first-instar nymphs are difficult to be tracked since they have well-developed legs, and the second-instar stage is the most suitable stage to get the most-synchronized nymphs for studying their difference in development duration.

Point 3: P248: Fig1 Maybe it will be better to list the sequences numbers of these target species.

Response 3: Thank you for your advice. As requested, we have listed all the accession numbers of sequences we used for phylogenetic analysis in Table S3.

Best wish!

Dr You-Jun Zhang, Professor of Entomology

Institute of Vegetables and Flowers

Chinese Academy of Agricultural Science

Beijing 100081; China

Tel:86-10-62152945; 82109518

Fax:86-10-82109518

Reviewer 2 Report

The authors have provided an analysis involving classification, phylogenetic analysis, gene expression studies, and gene silencing studies of chitinase family genes in Bemisia tabaci, for potential use as pest control in crops. The main idea is to utilize the 3 chitinases identified as significantly impeding the growth of nymphs, in nanomaterial based RNAi application. The language needs to be improved significantly before publication. In the paragraphs below, I have provided detailed suggestions and comments on the scientific content, and some on the language for improving this manuscript for publication.

Abstract:

Line 24: Please modify genome wide panorama to genome wide distribution, if that is the intention.

Line 26: change encoding to encode

Introduction

Line 65: change shedding off to shed off

Line 81-82: There seem to missing sentences here. Please reread and modify the statement.

Line 85: Remove “And” from the beginning of the sentence.

Line 88-93: Modify this section to

More chitinase sequences need to be analyzed as there are still chitinases in these hemimetabolous insects that could not be clustered into a certain group. From this view, the inadequate annotation data on chitinase-like genes of different insect species may account for the inability to classify certain genes. Hence the genome-wide identification of chitinase-like gene family in more insect species is necessary for accelerating the classification of insect chitinases.

Line 98: Change remains to remaining

Line 101: Change newish to new

Line 105: Change panoptic to genome wide

Line 106: Please provide information on Bemisia tabaci genome including references for this genome release.

Materials and Methods

Line 130: References are needed for the genome and transcriptome sequences used for mining the chitinases.

Line 132: Reference is needed for the PRIMER PREMIER 5 program used for desigining primers.

Line 134-135: PCR conditions need to be provided in a supplementary table for each primer used.

Line 162-164: GenBank accession numbers for each of the 14 genes used to study the exon intron structure needs to be provided in a supplemenatry table.

Line 181: Livak and Schmittgen reference needs to be formatted to match the journal refernecing guidelines.

Line 198: References are needed for the nanomaterial promoted RNAi method.

Further, the introduction section needs to address details about the nanomaterial promoted RNAi method, including its usage, efficiency, and any unintended transfer to other organisms as well as other consequences. In the introduction section, please also provide information on how such a program can be implemented.

Section 2.8: Provide references for both methods used.

Results

Line 362-365: Split this long sentence into 2. The first sentence should stop at “..higher in nymph than that of adult.”

Further, please clarify the next statement. Do you intend to say that the potential role of these gene sin immature mottling is implied but needs to be verified?

Line 366: Replace demonstarting with “may suggest”.

Figure 7: The description does not match the figure shown. Please add appropriate description.

Discussions:

Line 496: Change to: “The increasing availability of genomic data…”

Line 501-504: This statement is unclear. Please reword and clarify this statement.

Line 553-556: Change to “…along with previous research, enriches...”

Conclusions:

Please reread the conclusion and correct all grammatical and typographical errors.

Author Response

Dear reviewer:

Thanks for your patient revision and professional advice on our article. We have made point-by-point modifications based on the valuable comments you gave us and details are as follow:

Abstract:

Point 1: Line 24: Please modify genome wide panorama to genome wide distribution, if that is the intention.

Response 1: Thanks for your suggestion. We have made modification to this phrase in Line 24 as suggested.

Point 2: Line 26: change encoding to encode

Response 2: Thanks for your advice. We are sorry to make such error and have corrected in Line 26.

Introduction

Point 3: Line 65: change shedding off to shed off

Response 3: Thanks for your suggestion. We have rectified this error in Line 65.

Point 4: Line 81-82: There seem to missing sentences here. Please reread and modify the statement.

Response 4: Thanks for your attention. We are sorry to make confusions and have rewrite this sentence in Line 81-83.

Point 5: Line 85: Remove “And” from the beginning of the sentence.

Response 5: Thanks for your advice. “And” has been removed.

Point 6: Line 88-93: Modify this section to

More chitinase sequences need to be analyzed as there are still chitinases in these hemimetabolous insects that could not be clustered into a certain group. From this view, the inadequate annotation data on chitinase-like genes of different insect species may account for the inability to classify certain genes. Hence the genome-wide identification of chitinase-like gene family in more insect species is necessary for accelerating the classification of insect chitinases.

Response 6: Thanks for your nice revision. We have made modifications in Line 90-95 as you suggested.

Point 7:

Line 98:Change remains to remaining

Line 101: Change newish to new

Line 105: Change panoptic to genome wide

Response 7: Thanks for your appropriate suggestions. We have changed these three words in Line 122, Line 126 and Line 130.

Point 8: Line 106: Please provide information on Bemisia tabaci genome including references for this genome release.

Response 8: Thanks for your attention. We have added reference on information of the genome. (Line132)

Materials and Methods

Point 9:

Line 130: References are needed for the genome and transcriptome sequences used for mining the chitinases.

Line 132: Reference is needed for the PRIMER PREMIER 5 program used for desigining primers.

Response 9: Thanks for your kind reminding. We have added related references in Line 169 and Line 171.

Point 10: Line 134-135: PCR conditions need to be provided in a supplementary table for each primer used.

Response 10: Thanks for your advive. We have supplemented the PCR conditions in Table S2.

Point 11: Line 162-164: GenBank accession numbers for each of the 14 genes used to study the exon intron structure needs to be provided in a supplemenatry table.

Response 11: Thanks for your suggestion. We have supplemented the GenBank accession numbers in Table S3.

Point 12: Line 181: Livak and Schmittgen reference needs to be formatted to match the journal refernecing guidelines.

Response 12: Thanks for your attention. We have formatted the reference as the referencing guidelines requested.

Point 13: Line 198: References are needed for the nanomaterial promoted RNAi method.

Further, the introduction section needs to address details about the nanomaterial promoted RNAi method, including its usage, efficiency, and any unintended transfer to other organisms as well as other consequences. In the introduction section, please also provide information on how such a program can be implemented.

Response 13: Thanks for your advice. We have supplemented related informations about the nanomaterial and referenced several articles to address details about the method. (Line 102-109)

Point 14: Section 2.8: Provide references for both methods used.

Response 14: Thanks for your suggestion. References for both methods used have all been provided in Line 269 and Line 261.

Results

Point 15: Line 362-365: Split this long sentence into 2. The first sentence should stop at “..higher in nymph than that of adult.”

Further, please clarify the next statement. Do you intend to say that the potential role of these genes in immature mottling is implied but needs to be verified?

Response 15: Thanks for your thorough consideration. We have made the revisions as you suggested. And the intension of the next statement is exactly matched with your understandin. Regretfully, the word “implicit” made some expressional confussions and we have made modifications to this statement in Line 406.

Point 16: Line 366: Replace demonstrating with “may suggest”.

Response 16: Thanks for your attention. We have made the replacement as you suggested.

Point 17: Figure 7: The description does not match the figure shown. Please add appropriate description.

Response 17: Thanks for your reminding and we are sorry for such mistake. We have rectified the description which now matches Figure 7.

Discussions:

Point 18: Line 496: Change to: “The increasing availability of genomic data…”

Response 18: Thanks for your advice. We have made changes to this sentence as suggested.

Point 19: Line 501-504: This statement is unclear. Please reword and clarify this statement.

Response 19: We are sorry to make such confussion and have reworded the sentence. (Line 545-547) 

Point 20: Line 553-556: Change to “…along with previous research, enriches...”

Response 20: Thanks for your suggestion. We have made changes to this sentence in Line 596-597.

Conclusions:

Point 21: Please reread the conclusion and correct all grammatical and typographical errors.

Response 21: Thanks for your kind reminding. We have made inspections to the conclusions part and corrected all grammatical and typographical errors.

Best wish!

Dr You-Jun Zhang, Professor of Entomology

Institute of Vegetables and Flowers

Chinese Academy of Agricultural Science

Beijing 100081; China

Tel:86-10-62152945; 82109518

Fax:86-10-82109518

Reviewer 3 Report

The manuscript by Pen et al identified and classified members of the chitinase family in B. tabaci MED, and placed them into context with other insect species.  Following expression analysis across lifecycle stages, the group used RNAi to knock down expression of selected chitinase genes, which resulted in increased mortality and reduced survival of a molt course.  The figures, particularly figure 6, are well presented, and the writing is comprehensive.

Some points:

It would be worthy in the introduction to discuss the differences between MEAM1 and MED biotypes, as this paper focusses on MED.  There are previously annotated chitinase genes in the publically-available MEAM1 genome, which should be referenced.  The previous use of dietary RNAi to for induce mortality and study gene function should be introduced and referenced (there are examples for other species for chitinase even).   

The materials and methods are lacking in some areas.  What is the nanomaterial used for the nanomaterial-RNAi?  The order also does not seem correct; cDNA synthesis and cloning is shown before gene identification, when it should be after.

The function of the nanomaterial and why it was used should be discussed and referenced.  

The Figure 1 legend needs fixing.  Additionally, grammar and spelling errors are relatively common.  I.e. function genomics (71);  And the typical hemimetabolous insect, Nilaparvata lugens, have also been conducted RNA interference experiments (81), etc.

Author Response

Dear reviewer:

Thanks for your patient revision and professional advice on our article. We have made point-by-point modifications based on the valuable comments you gave us and details are as follow:

Point 1: It would be worthy in the introduction to discuss the differences between MEAM1 and MED biotypes, as this paper focusses on MED.  There are previously annotated chitinase genes in the publically-available MEAM1 genome, which should be referenced.  The previous use of dietary RNAi to for induce mortality and study gene function should be introduced and referenced (there are examples for other species for chitinase even).

Response 1: Thanks for your advices. We have introduced the differences between MEAM1 and MED biotypes in Line113-121, which we also think it necessary since we focused on MED. Also, the reference about chitinase genes in MEAM1 genome have been added in Line 129. Informations and references about the application of RNAi to study gene function have been added in Line 102-109 as well.

Point 2: The materials and methods are lacking in some areas.  What is the nanomaterial used for the nanomaterial-RNAi?  The order also does not seem correct; cDNA synthesis and cloning is shown before gene identification, when it should be after.

Response 2: Thanks for your thorough considerations. We have supplemented informations about the nanomaterial-RNAi in Line 102-109. The nanomaterial was functioned as a carrier that promotes the transdermal delivery of dsRNA into insects, and details about the material have been added and referenced. The order in material and methods has been corrected.

Point 3: The function of the nanomaterial and why it was used should be discussed and referenced.

Response 3: Thanks for your thorough considerations. The function of the nanomaterial has been discussed and referenced in Line102-109.

Point 4: The Figure 1 legend needs fixing.  Additionally, grammar and spelling errors are relatively common.  I.e. function genomics (71);  And the typical hemimetabolous insect, Nilaparvata lugens, have also been conducted RNA interference experiments (81), etc.

Response 4: Thanks for your suggestions. We are sorry about the mistake in the Figure 1 legend and we have corrected. We have also corrected the grammar and spelling errors as well.

Best wish!

Dr You-Jun Zhang, Professor of Entomology

Institute of Vegetables and Flowers

Chinese Academy of Agricultural Science

Beijing 100081; China

Tel:86-10-62152945; 82109518

Fax:86-10-82109518

Round 2

Reviewer 2 Report

The manuscript needs to provide more details on the safety of RNAi based method of controlling white flies on non targets. Please provide information on the safety metrics from the references that you provided in the introduction (Line 106-109, new manuscript).

Author Response

Dear reviewer:

Thanks for your patient revision and professional advice on our article. We have made point-by-point modifications based on the valuable comments you gave us and details are as follow:

Point 1: The manuscript needs to provide more details on the safety of RNAi based method of controlling whiteflies on non-targets. Please provide information on the safety metrics from the references that you provided in the introduction (Line 106-109, new manuscript).

Response 1: Thank you for your good advice. As you suggested, we have supplemented detailed information in Line 107-114.

“It is always necessary to concern for the safety of non-targets when RNAi-based method of controlling pests is to be applied in the field. As we know, dsRNA can specifically target gene transcript without affecting non‐target species, which on the one hand ensures the safety of non-targets. On the other hand, as research showed that silencing of the aphid hemocytin gene (Hem) by a nanocarrier-promoted RNAi method dramatically decreased aphid population density, while the cell survival ratios of S2 cells with the same treatment was more than 96%, which demonstrates the safety of the RNAi method to non-targets [18].”

Best wish!

Dr You-Jun Zhang, Professor of Entomology

Institute of Vegetables and Flowers

Chinese Academy of Agricultural Science

Beijing 100081; China

Tel:86-10-62152945; 82109518

Fax:86-10-82109518
